# From Homeostasis to Pathology: Decoding the Multifaceted Impact of Aquaporins in the Central Nervous System

**DOI:** 10.3390/ijms241814340

**Published:** 2023-09-20

**Authors:** Corneliu Toader, Calin Petru Tataru, Ioan-Alexandru Florian, Razvan-Adrian Covache-Busuioc, David-Ioan Dumitrascu, Luca Andrei Glavan, Horia Petre Costin, Bogdan-Gabriel Bratu, Alexandru Vlad Ciurea

**Affiliations:** 1Department of Neurosurgery, “Carol Davila” University of Medicine and Pharmacy, 020021 Bucharest, Romania; corneliu.toader@umfcd.ro (C.T.); razvan-adrian.covache-busuioc0720@stud.umfcd.ro (R.-A.C.-B.); david-ioan.dumitrascu0720@stud.umfcd.ro (D.-I.D.); luca-andrei.glavan0720@stud.umfcd.ro (L.A.G.); horia-petre.costin0720@stud.umfcd.ro (H.P.C.); bogdan.bratu@stud.umfcd.ro (B.-G.B.); prof.avciurea@gmail.com (A.V.C.); 2Department of Vascular Neurosurgery, National Institute of Neurology and Neurovascular Diseases, 077160 Bucharest, Romania; 3Department of Opthamology, “Carol Davila” University of Medicine and Pharmacy, 020021 Bucharest, Romania; 4Central Military Emergency Hospital “Dr. Carol Davila”, 010825 Bucharest, Romania; 5Department of Neurosciences, “Iuliu Hatieganu” University of Medicine and Pharmacy, 400012 Cluj-Napoca, Romania; 6Neurosurgery Department, Sanador Clinical Hospital, 010991 Bucharest, Romania

**Keywords:** aquaporins, central nervous system, fluid homeostasis, glymphatic system, neurodegenerative diseases, brain tumorigenesis, permeability assays, AQP4, AQP1, stroke, Parkinson’s disease, Alzheimer’s disease, neuromyelitis optica, cellular physiology, therapeutic interventions

## Abstract

Aquaporins (AQPs), integral membrane proteins facilitating selective water and solute transport across cell membranes, have been the focus of extensive research over the past few decades. Particularly noteworthy is their role in maintaining cellular homeostasis and fluid balance in neural compartments, as dysregulated AQP expression is implicated in various degenerative and acute brain pathologies. This article provides an exhaustive review on the evolutionary history, molecular classification, and physiological relevance of aquaporins, emphasizing their significance in the central nervous system (CNS). The paper journeys through the early studies of water transport to the groundbreaking discovery of Aquaporin 1, charting the molecular intricacies that make AQPs unique. It delves into AQP distribution in mammalian systems, detailing their selective permeability through permeability assays. The article provides an in-depth exploration of AQP4 and AQP1 in the brain, examining their contribution to fluid homeostasis. Furthermore, it elucidates the interplay between AQPs and the glymphatic system, a critical framework for waste clearance and fluid balance in the brain. The dysregulation of AQP-mediated processes in this system hints at a strong association with neurodegenerative disorders such as Parkinson’s Disease, idiopathic normal pressure hydrocephalus, and Alzheimer’s Disease. This relationship is further explored in the context of acute cerebral events such as stroke and autoimmune conditions such as neuromyelitis optica (NMO). Moreover, the article scrutinizes AQPs at the intersection of oncology and neurology, exploring their role in tumorigenesis, cell migration, invasiveness, and angiogenesis. Lastly, the article outlines emerging aquaporin-targeted therapies, offering a glimpse into future directions in combatting CNS malignancies and neurodegenerative diseases.

## 1. Introduction

### 1.1. Definition and General Characteristics of Aquaporins

Aquaporins represent a ubiquitous family of integral membrane proteins that are disseminated extensively across both animal and plant kingdoms. Structurally, these proteins are characterized by a core architecture comprising six alpha-helical transmembrane domains and two additional, shorter helical elements. These components delineate cytoplasmic and extracellular vestibules, which are connected via a constricted aqueous pore (Figure 1). Notably, conserved sequence motifs, such as the asparagine-proline-alanine (NPA) motifs, are frequently observed within these shorter helical regions. Functionally, aquaporin monomers autonomously integrate into the lipid bilayer, subsequently aggregating as tetrameric complexes. Certain variants, exemplified by mammalian Aquaporin 4 (AQP4), have the capability to further organize into higher-order structures, known as orthogonal arrays of particles, within the cell membrane [1,2].

The predominant role of most aquaporins is to facilitate the translocation of water molecules across cellular membranes in reaction to osmotic gradients, which are typically established via active solute transport mechanisms. Owing to the relatively low individual throughput of water by aquaporin monomers, membranes are often saturated with a high density of these proteins—up to 10,000 per square micron—to substantially augment the membrane’s overall water permeability [3]. Computational studies, particularly molecular dynamics simulations, posit that the specificity of aquaporins for water is mediated by steric constraints and electrostatic interactions within the channel’s aqueous pore [4].

Additionally, a specialized subgroup of aquaporins, termed aquaglyceroporins, exhibits the capacity to transport glycerol molecules. The structural configuration of aquaglyceroporins is distinguished by a marginally larger pore diameter and a lining enriched with hydrophobic amino acid residues, in contrast to the hydrophilic nature of the pore in water-selective aquaporins. Beyond the primary functions of transporting water and, in some cases, glycerol, aquaporins are postulated to facilitate the passage of various gases (CO_2_, NH_3_, NO, O_2_) and small solutes such as hydrogen peroxide and arsenite. Some aquaporins have even been implicated in ion transport (K^+^, Cl^−^), although these claims remain a subject of ongoing debate [5].

Moreover, a range of non-transport-centric functions have been ascribed to some aquaporins, encompassing roles in cellular adhesion, membrane polarization, and the modulatory regulation of protein interactions, such as those involving ion channels [5].

### 1.2. The Overarching Significance of Aquaporins in Cellular Physiology, with an Emphasis on Their Role in the Central Nervous System

The expression profiles of Aquaporins (AQPs), particularly AQP4 and AQP1, have garnered significant research attention within the context of the central nervous system and sensory organs, as compared to the peripheral and enteric nervous systems [6]. AQP4, the predominant water channel in the brain, spinal cord, and optic nerve, is largely expressed in the astrocyte cell plasma membrane but is distinctly localized to specialized regions such as astrocyte foot processes [7,8,9]. Such polarized expression patterns are hypothesized to be mediated by intracellular associations between AQP4 and α-syntrophin or through extracellular interactions with agrin. Both α-syntrophin and agrin also exhibit polarized expression in astrocyte foot processes [10].

Additionally, the formation of orthogonal arrays of particles (OAPs) by AQP4 may facilitate its polarized localization, as it requires only a single anchoring link for stabilization, as opposed to individual AQP4 tetramers [10]. In the CNS, AQP4 is predominantly localized to the subpial astrocyte processes forming the glial-limiting membrane, the perivascular astrocyte endfeet, and the basolateral membrane of ependymal and subependymal astrocyte processes [11,12]. Such strategic localization suggests that AQP4 may play a critical role in regulating water flux across the CNS-water compartment interfaces.

In specific brain regions lacking a blood–brain barrier, such as the circumventricular organs, and in hippocampal regions such as CA1 and the dentate gyrus, AQP4 manifests a more diffused expression profile [13]. This dispersed distribution may facilitate rapid water flux essential for maintaining potassium (K^+^) homeostasis during neuronal activity.

AQP1, on the other hand, is prominently found in the ventricular-facing plasma membrane of choroid plexus epithelial cells, implicating its role in cerebrospinal fluid (CSF) secretion [14]. While AQP1 is ubiquitously expressed in vascular endothelial cells throughout the body, it is conspicuously absent in the cerebrovascular endothelium, except in areas such as the circumventricular organs. Intriguingly, astrocyte co-culture leads to the suppression of AQP1 mRNA expression in primary brain microvessel endothelial cells, suggesting that astrocyte-endothelial interactions may down-regulate endothelial AQP1 expression [15].

The expression of AQP9 within the brain is relatively sparse, and its precise localization remains ambiguous due to the limitations in antibody specificity [16]. However, some evidence points to its expression in specialized neural cells such as neurons in the substantia nigra, tanycytes, and a subset of astrocytes.

In the spinal cord, the expression profiles of various AQPs are notably differentiated. AQP4 is predominantly localized to perivascular astrocyte foot processes and the glial-limiting membrane, whereas AQP1 is expressed in the processes of non-myelinated neurons in the dorsal horns. AQP9 is purportedly expressed in spinal cord radial astrocytes and also potentially in the glial-limiting membrane [16].

Similar to its expression patterns in the brain and spinal cord, AQP4 is primarily found in perivascular astrocyte foot processes and the glial-limiting membrane within the optic nerve [7]. Additionally, AQP4 expression extends to astrocyte processes in CNS regions lacking a blood-brain barrier, including but not limited to the pre-laminar optic nerve head, circumventricular organs, and root entry zones in the spinal cord. This peripheral exposure of the CNS AQP4 pool suggests its potential susceptibility as an initial target for circulating AQP4-specific antibodies, particularly in the context of neuromyelitis optica (NMO), a topic warranting further exploration [3].

## 2. Historical Perspective on Aquaporins

The scientific exploration of water transport mechanisms predates the molecular characterization of aquaporins, which initially focused on tissues with notable water permeability. The seminal discovery of the first aquaporin, AQP1, was made during research endeavors aimed at identifying the Rh blood group antigens. Since this initial discovery, the field has undergone substantial expansion, extending the study of aquaporins to diverse organisms. In mammals, the aquaporin family exhibits functional heterogeneity. While certain isoforms specialize exclusively in facilitating water transport, others are capable of translocating a diverse array of solutes across cellular membranes. Regulatory mechanisms also exist that modulate aquaporin function, either by influencing channel permeability or by altering subcellular localization. Despite extensive research elucidating the physiological roles of various mammalian aquaporins, significant knowledge gaps remain within this domain. Understanding the comprehensive physiological implications of this integral membrane protein family is still evolving and presents an area ripe for further scholarly inquiry [17].

### 2.1. Early Studies of Water Transport

Initial investigations into water transport mechanisms commenced with the observation that certain amphibian tissues, such as the skin and bladder—which bear functional resemblance to the mammalian kidney’s collecting duct—exhibited enhanced water permeability compared to other tissues. Early work by Hans Ussing and collaborators highlighted the particular water permeability of amphibian skin [18]. Subsequent electron microscopic analyses identified structural assemblies in amphibian bladder tissues, presumed to be water channels, whose prevalence correlated with increased tissue water permeability [19].

The concept of regulated water permeability through localized water channels was initially postulated under the ‘shuttle hypothesis’. In this framework, protein aggregates were detected in intracellular vesicles during periods of low water reabsorption (diuresis) and were translocated to the plasma membrane during periods of increased water reabsorption (antidiuresis) [20].

Moreover, red blood cells were recognized for their high water permeability. Investigations by A.K. Solomon and associates revealed water transport in red blood cells that exhibited low Arrhenius activation energy, implying the existence of membrane pores facilitating the water movement [21]. Further studies by Robert Macey and colleagues indicated that the water permeability of red blood cells could be chemically modulated. Specifically, water transport inhibition via HgCl_2_ could be reversed by chemical reducing agents, suggesting the presence of a protein featuring accessible free sulfhydryl groups sensitive to mercury [22].

In the quest to ascertain the molecular identity of water channels [23], a myriad of strategies were employed by different research groups. Among these efforts were the radiolabeling studies conducted by William Harris and colleagues, who compared proteins from toad bladders exposed to antidiuretic hormones (ADH) with those unexposed, seeking to identify proteins localized to the plasma membrane in response to ADH [24]. Other approaches included expression cloning utilizing mRNA from tissues with known water transport properties and radiation inactivation techniques aiming to ascertain the molecular weight of water channels [25]. Despite these extensive efforts, the molecular identity of aquaporins remained elusive through these methodologies.

### 2.2. Discovery of Aquaporin 1

In a groundbreaking development for the understanding of water channels, Benga’s group in 1985 located a water channel protein among the polypeptides in the 35–60 kDa range on the electrophoretogram of red blood cell (RBC) membrane proteins. Subsequent work by Agre and colleagues in 1988 isolated a novel 28-kDa protein from the RBC membrane, termed CHIP28 (channel-forming integral membrane protein of 28 kDa). Intriguingly, this protein also had a glycosylated component in the 35–60 kDa range, aligning with Benga’s findings. It was not until 1992 that Agre’s team posited that CHIP28 likely represented a functional membrane water channel. The seminal contributions of Benga’s group from 1986 were largely overlooked when Peter Agre was awarded half of the 2003 Nobel Prize in Chemistry for the “discovery of water channels”. This omission sparked widespread support for Benga in the scientific community [26].

The discovery of the first water channel, AQP1, was rooted in a search for Rh blood group antigens [27]. A 32-kDa protein related to one of the Rh antigens was isolated using hydroxylapatite chromatography. Interestingly, the protein did not stain well with Coomassie stain. However, upon using silver reagent, a separate 28-kDa protein was revealed. Initially thought to be a fragment of the 32-kDa protein, an antibody test disproved this hypothesis, confirming that the 28-kDa protein was unique and had properties akin to a membrane channel [28]. This protein was temporarily named CHIP28 [29]. John C. Parker, a membrane physiologist, was the first to suggest that CHIP28 might be the elusive water channel, based on its presence in red blood cells and specific renal tissues.

Further characterization led to the cloning of the cDNA encoding CHIP28, which was found to share sequence similarity with uncharacterized genes from microbes and plants [30]. Definitive evidence for CHIP28 as a water channel was obtained through its expression in Xenopus laevis oocytes, a model organism that had been previously used to study other transport proteins [31]. This marked a major milestone in the scientific understanding of membrane water channels, albeit with a backdrop of controversy regarding the acknowledgment of pioneering work.

Erich Windhager, based at Cornell Medical School, was the first to propose using oocytes for studying water transport, given their naturally low water permeability. Experiments using Xenopus laevis oocytes expressing CHIP28 presented compelling results. When placed in a dilute Modified Barth’s solution with a lowered osmolarity (70 mosM as opposed to the standard 200 mosM), these oocytes swelled rapidly and burst, contrasting sharply with control oocytes, which exhibited negligible swelling [32].

The water permeability coefficient (Pf) of the CHIP28-expressing oocytes was found to be 20 times higher than that of the controls. The Arrhenius activation energy of these specialized oocytes was also lower, aligning with previous observations in native membranes. Importantly, the addition of 1 mM HgCl2 inhibited water permeability in CHIP28 oocytes, an effect reversible by a reducing agent. This was consistent with earlier studies on water-permeable tissues, reinforcing the idea that CHIP28 was instrumental in water transport [22].

To discount the possibility that CHIP28 merely activated an endogenous water channel in oocytes, purified CHIP28 was integrated into membrane proteoliposomes. The result was striking: the water permeability of these proteoliposomes increased by 50-fold, while urea and proton transport remained unaffected. It was estimated that a single CHIP28 subunit could facilitate the passage of roughly 3 × 10^9^ water molecules per second.

As researchers discovered more homologs of CHIP28, the term “aquaporin” was introduced to denote this expanding family of water channels [33]. Accordingly, the original CHIP28 was renamed Aquaporin 1 (AQP1).

AQP1 serves as a model for understanding the aquaporin family, which spans across microorganisms, plants, and mammals. Early studies using red blood cells suggested that AQP1 exists as a homotetramer in the membrane. Each subunit of the tetramer is mostly embedded in the membrane, with the N- and C-termini of the polypeptide chain extending into the cytoplasm [29]. Negative staining electron microscopy of cells expressing AQP1 and proteoliposomes confirmed this tetrameric structure [34].

Prior work on the lens MIP protein, now identified as AQP0, had proposed that there were six domains of the protein spanning the membrane [35]. This was later confirmed for AQP1 through experiments that involved inserting an epitope tag at varying points along the polypeptide chain. The location of the epitope within the membrane was ascertained via proteolysis and antibody targeting. The experimental findings helped elucidate the architecture of the helices in the membrane, revealing them to be obversely symmetric. This led to the proposal that AQP1 features two highly conserved NPA (asparagine-proline-alanine) motifs located adjacently in the lipid bilayer, giving the protein an hourglass-like structure [36].

This model of AQP1’s structure has been further validated by high-resolution studies of AQP1 and other members of the aquaporin family. The research has significantly advanced our understanding of how these proteins function at a molecular level, illuminating their role in facilitating water transport across cell membranes.

## 3. Distribution and Molecular Classification of Aquaporins

### 3.1. Overview of Various Aquaporin Isoforms Present in Mammalian Systems

In mammals, the aquaporin (AQP) family is generally made up of 12 to 15 isoforms grouped into 13 subfamilies (AQP0–12). However, humans stand out by having 18 paralogs due to tandem duplications, including four extra AQP7 pseudogenes and a second copy of AQP12. Additionally, more ancient mammalian lineages, such as Metatheria and Prototheria, have been shown to possess further classes such as AQP13–14 [37].

Animal AQPs are typically categorized into four main groups:Classical or Orthodox AQPs (AQP0, 1, 2, 4, 5, 6, 14), which are mainly focused on water transport.Aqua-ammoniaporins (AQP8), which are sometimes counted among the orthodox AQPs.Aquaglyceroporins (AQP3, 7, 9, 10, 13), which can transport glycerol in addition to water.Unorthodox AQPs (AQP11–12), also known as “superaquaporins”, which have low sequence homology with other AQPs and feature unusual N-terminal NPA motifs [37,38,39,40].

Plants show even more diversity in their aquaporin families, likely due to higher genome duplication rates and their stationary lifestyles requiring more specialized water transport mechanisms. For example, the model plant species Arabidopsis thaliana has 35 isoforms, Zea mays has 38, and upland cotton (*Gossypium hirsutun*) has 71. The current record holder is oil seed rape (*Brassica napus*), with 121 full-length AQPs [41,42,43].

Plant AQPs are typically classified into seven subfamilies based on subcellular localization and other specific features:Plasma Membrane Intrinsic Proteins (PIPs);Tonoplast Intrinsic Proteins (TIPs);NOD26-Like Intrinsic Proteins (NIPs);Small Basic Intrinsic Proteins (SIPs);Unknown Intrinsic Proteins (XIPs), which are absent in monocots and Brassicaceae;Hybrid Intrinsic Proteins (HIPs) and GLpF-like intrinsic proteins (GIPs), which are found only in older plant lineages such as mosses [44,45,46].

The origins and evolutionary roles of these plant aquaporin families remain subjects of ongoing research. For instance, it is still debated whether NIPs are ancestral or acquired through horizontal gene transfer [37,38,39].

Green algae also possess a unique set of AQPs, including PIPs and GIPs, along with five other families (MIP A–E) not seen in plants [47]. A completely unique family, large intrinsic proteins (LIPs), was recently discovered in diatoms [48]. These various subfamilies can further be divided into paralog groups, enhancing the complexity and adaptability of water transport across different organisms.

The plant intrinsic protein (PIP) subfamily in plants is usually divided into PIP1 and PIP2 groups based on sequence homology [49]. These divisions are functionally significant, as the two groups have distinct capabilities in terms of membrane localization and water transport activity. In terms of nomenclature, plant major intrinsic proteins (MIPs) are usually named based on the abbreviation of their subfamily, along with two numbers denoting their phylogenetic group and order of discovery. However, this system of naming can be confusing, especially when comparing proteins across species. Two proteins with identical names from different species may not actually be orthologous [39].

Recent phylogenetic research has refined our understanding further, identifying 19 clusters of orthologous genes in flowering plants. For example, the PIP family, previously divided into two clusters, was found to be divided into three, aligning with prior classifications in Arabidopsis thaliana [38]. This has led to calls for a standardized, globally accepted nomenclature based on evolutionary relationships. Such a nomenclature could facilitate the comparison and integration of knowledge across both animal and plant MIPs.

A particular phylogenetic study highlighted the close evolutionary relationships between animal and plant MIP subfamilies when considering proteins with a high amino acid identity (>25%). The study suggested a model of vertical transfer for four ancestral subfamilies [38]. In this model, AQP1-like and PIP subfamilies are grouped together as family A; AQP8-like and TIP as family B; AQP3-like and NIP as family C; and AQP11-like and SIP as family D. This grouping is not merely theoretical but also has practical implications. For instance, it can be used to predict substrate specificities across different MIP subfamilies.

### 3.2. The Molecular Underpinnings of Their Selective Permeability and Functionality

Water transport assays to study Aquaporins have been conducted using a variety of model systems across species, including bacteria, yeast, and mammalian cells [50,51,52]. This diversity in experimental setups reflects the widespread distribution of AQPs in nature. Specifically, cellular components such as intracellular and plasma membrane vesicles, particularly from animal tissues such as the kidney and intestinal epithelia, have been essential for evaluating AQP activity [53].

One of the prominent methods for characterizing new AQP isoforms involves the use of Xenopus laevis oocytes, owing to their inherently low water permeability [32]. This heterologous expression system allows for functional studies of AQPs from various organisms. Yeast cells lacking endogenous AQPs also serve as an effective model system for this purpose. They have been used to develop a high-throughput assay to identify functionally relevant AQP mutants that can withstand freeze-thaw cycles. Additionally, AQP-transfected cell lines and Zebrafish embryos injected with AQP mRNA have been deployed for similar studies [54].

Research has not been limited to functional assays alone; purified AQPs have been reconstituted into liposomes to directly investigate their roles in water and solute transport. Moreover, the development of transgenic mice models lacking specific AQPs has enriched our understanding of these proteins [55]. Such mouse models have illuminated the multi-faceted roles of AQPs in various biological functions. These range from transepithelial fluid transport and cell migration to more complex physiological processes such as brain edema, neuroexcitation, cell proliferation, skin hydration, and even metabolic activities in adipocytes.

### 3.3. Permeability Assays

The quantification of water or solute permeability across biological membranes serves as an indirect method to assess both the expression and functional status of Aquaporins. In general, the analytical focus for AQP activity and water permeability revolves around tracking alterations in cell or vesicle volume, which result from osmotically or pressure-driven water fluxes [56]. For solute permeability estimations, it is essential to account for both solute and water fluxes when examining volume changes, as these fluxes are instigated by their respective gradients.

Both water and solute fluxes are directly correlated to their inducing forces, represented by osmotic permeability (Pf) and solute permeability (Ps) coefficients, respectively. The kinetics of volume changes depend on the partitioning of water or solute between the aqueous pathway through the channel and diffusion across the lipid bilayer. Furthermore, analyzing permeability as a function of temperature enables the calculation of the activation energy (Ea) required for transport, a crucial metric for gauging AQP functionality. It is generally observed that water or solute fluxes via hydrophilic channel pores necessitate lower activation energy compared to fluxes traversing a hydrophobic lipid bilayer. Consequently, high permeability coupled with low Ea is indicative of AQP-mediated transport [57].

Analytical methodologies for assessing permeability commonly employ optical properties that are volume-dependent, such as light transmission, absorbance, scattering, and fluorescence [58,59,60]. These strategies are equally applicable to evaluating transport kinetics in both cellular models and proteoliposomes. The amalgamation of diverse biological models (cells, vesicles, and proteoliposomes) with optical detection systems offers a comprehensive toolkit for the investigation of AQP functions.

### 3.4. Epithelial Assays

Water permeability can be assessed in native epithelial tissues, such as those from the intestinal wall or kidney tubules, as well as in cultured epithelial cell monolayers positioned on permeable supports within Ussing chambers [61]. In these experiments, the apical and basolateral membranes of polarized cells are exposed to distinct compartments. The introduction of a membrane-impermeable solute such as sucrose or mannitol to one of these compartments instigates a transepithelial water flux. This flux is then quantified either by observing the fluid level change in a capillary tube linked to the opposite compartment or through the use of a fluorescent dye. The dye is added to the hyperosmotic compartment, and the rate of fluorescence alteration due to dye dilution serves as a measure of overall transepithelial osmotic water permeability [62].

It is crucial to note that the total transepithelial permeability is the aggregate of two distinct pathways: the cellular and the paracellular pathways. In this context, the presence of AQPs specifically influences the cellular pathway. To assess the permeability of individual membranes in these bipolar epithelial cells, isolated vesicles from either the basolateral or apical membranes can be employed. This methodology is prevalently utilized for both the identification and functional characterization of AQPs in epithelial membranes.

### 3.5. Osmotic Swelling Assays

The functionality of AQP-mediated water transport can be probed using X. laevis oocytes in an osmotic swelling assay [56]. Oocytes that are microinjected with AQP mRNA are exposed to hypo-osmotic conditions, and the kinetics of cellular swelling are monitored using video microscopy. To investigate solute permeability, an inwardly directed chemical gradient is established, prompting a solute influx followed by an influx of water, culminating in oocyte swelling [63]. The low intrinsic water permeability and negligible permeability for glycerol and other solutes in oocytes make this system particularly well-suited for AQP studies.

An analogous swelling assay employs erythrocytes, which express native AQPs. In humans, erythrocytes predominantly express a single aquaglyceroporin isoform, AQP3. When subjected to hyperosmotic solute gradients, such as glycerol, the ensuing influx of glycerol provokes erythrocyte swelling, eventually leading to cell hemolysis. This hemolysis can be tracked as a reduction in light absorbance at 625 nm [64]. The rate constant derived from the hemolytic process can be employed to determine glycerol permeability.

## 4. Functional Dynamics of Aquaporins in the Brain

### 4.1. Detailed Exploration of AQP4 and AQP1, Emphasizing Their Role in Maintaining Fluid Equilibrium across Neural Compartments

Aquaporin 1, localized in the choroid plexus epithelial cells, and Aquaporin 4, found in ependymal cells as well as glial limitants, are implicated in the regulation of cerebrospinal fluid homeostasis and production. The specific roles played by these individual water channels in these processes are currently under academic debate. Evidence suggests that both AQP1 and AQP4 contribute significantly to CSF production and that a concurrent mutation in both AQP1 and AQP4 genes disrupts CSF drainage and ventricular compliance. Data further underscore the role of AQP4 in extra-choroidal CSF formation, advocating for a critical and sustained balance in CSF production and absorption, facilitated by water flux between brain capillaries and interstitial fluid (ISF). Additionally, findings indicate that AQPs also participate in structural capacities related to CSF homeostasis, including the distensibility of the ventricular system [65].

Cerebrospinal fluid and interstitial fluid form integral parts of the cerebral extracellular milieu, existing in a dynamic equilibrium that bathes neurons and glial cells. This fluid balance, maintained through a tightly regulated exchange between CSF and ISF, is essential for ensuring stable brain volume as well as ionic and solute concentrations for optimal neural and glial function [66,67,68]. Traditionally, CSF, which is synthesized from plasma modification, has been understood to occupy the brain ventricles and subarachnoid spaces and serve primarily as a mechanical cushion for the central nervous system (CNS). It also provides a route for waste removal from neural tissues. According to the “classic view”, the choroid plexus is the principal site for CSF production, with fluid circulating unidirectionally and draining via arachnoid granulations or through the nasal mucosa’s lymphatic vessels [66,69].

However, recent advancements have challenged this “classic view”, proposing an updated paradigm wherein: (i) extra-choroidal CSF production occurs through fluid exchange between brain capillaries and ISF; (ii) there exists noteworthy paravascular CSF/ISF flow within the brain parenchyma; and (iii) the meningeal lymphatic system plays a significant role in CSF drainage [68,70,71]. Aberrations in CSF homeostasis are strongly correlated with the pathophysiology of various CNS conditions, including hydrocephalus, cerebral edema, and ischemia. Emerging evidence further indicates that diminished CSF production or impaired drainage may contribute to the decline in brain function associated with aging or age-related neurodegenerative disorders [72,73].

Cerebral aquaporins, specifically AQP1 and AQP4, are posited to be crucial regulators of cerebrospinal fluid (CSF) and interstitial fluid (ISF) homeostasis. Given their expression profiles—AQP1 localized solely in the choroid plexus epithelial cells and AQP4 in ependymal cells as well as glial limitants—it has been simplistically theorized that AQP1 is primarily involved in CSF production while AQP4 facilitates CSF/ISF exchange and absorption [74]. Consistent with traditional theories of CSF circulation, AQP1-deficient mice exhibited reduced CSF production and intraventricular pressure, underscoring the importance of AQP1 in CSF genesis [75].

However, a study involving AQP1-null and AQP4-null mice concluded that AQP4, rather than AQP1, is instrumental in mediating water influx into the CSF, suggesting a pivotal role for AQP4 in CSF production [76]. This finding lends credence to the “Bulat–Klarica–Oreskovic hypothesis”, which argues for widespread, continuous CSF formation in the brain due to water exchange between brain capillaries and ISF [70]. Moreover, the discovery of the so-called “glymphatic” system by Iliff et al., which facilitates AQP4-dependent paravascular flow between peri-arterial and peri-venous regions, has added further support to the crucial role of AQP4 in regulating CSF homeostasis [71].

Although the specific role of AQP4 within the “glymphatic” system remains a subject of ongoing discussion, there is general agreement that both AQP1 and AQP4 are indispensable for the regulation of cerebral fluid dynamics [67]. However, the exact contributions of AQP1 and AQP4 to various facets of CSF/ISF homeostasis—including CSF production and drainage, CSF/ISF interchange, and ventricular system regulation—remain to be elucidated.

### 4.2. Deep Dive into the Interplay between Aquaporins, Cerebrospinal Fluid, and the Intricacies of Brain Fluid Homeostasis

Aquaporin 1 (AQP1) and Aquaporin 4 (AQP4) serve as key water channel proteins in the central nervous system, each exhibiting unique but complementary expression patterns within cerebral tissues. Both are postulated to be central regulators of cerebral fluid homeostasis under both physiological and pathological conditions. Specifically, AQP1 is localized solely to the apical membranes of choroid plexus epithelial cells, while AQP4 is predominantly expressed in astrocytes and ependymal cells [6].

Traditionally, the choroid plexus has been viewed as the primary site for cerebrospinal fluid (CSF) production, facilitated by AQP1, with the fluid then flowing unidirectionally into the subarachnoid space for eventual reabsorption via arachnoid granulations. However, emerging evidence underscores the significance of extra-choroidal CSF formation, whereby CSF is generated through water filtration from brain parenchymal capillaries, facilitated by AQP4 [76].

In a study that performed in vivo analyses on wild-type, AQP1-null, AQP4-null, and double AQP1-AQP4-null mice to elucidate the specific contributions of AQP1 and AQP4 to critical aspects of CSF homeostasis, such as ventricular volume, CSF outflow dynamics, and ventricular compliance, Magnetic Resonance Imaging (MRI) and Intraventricular Pressure (IVP) measurements indicated that both AQP1 and AQP4 are implicated in CSF production. Interestingly, AQP1-null and AQP4-null mice exhibited comparable reductions in both ventricular volume and IVP, suggesting that choroidal AQP1 and parenchymal AQP4 contribute similarly to CSF formation. Consistent with this, double AQP-null mice displayed further reductions in ventricular volume compared to either single AQP1-null or AQP4-null mutants. Notably, despite the observed reduction in ventricular volume, double AQP-null mice maintained IVP levels comparable to those in single AQP-null mutants. These findings suggest a functional synergy between CSF production, whether facilitated by choroidal AQP1 or astroglial and ependymal AQP4, and ventricular volume to preserve intracranial pressure within homeostatic ranges [65].

The study sheds light on the role of Aquaporin 1 (AQP1) and Aquaporin 4 (AQP4) in cerebrospinal fluid (CSF) drainage dynamics. According to pressure-dependent outflow assessments, neither AQP1-null nor AQP4-null mutants exhibited significant alterations in CSF drainage, corroborating earlier findings from Verkman’s group [72]. Interestingly, data from double AQP-null mutants suggest that both channels could play a role, either directly or indirectly, in regulating CSF drainage dynamics. However, further research is needed to pinpoint where and how AQP1 and AQP4 contribute to this process. Some evidence of their involvement comes from hydrocephalus models, but the exact mechanisms remain elusive [77,78]. The recently described AQP4-mediated “glymphatic system” and its interaction with meningeal lymphatics provide additional support for the role of cerebral AQPs in interstitial fluid (ISF)/CSF exchange and drainage [68,71].

Regarding ventricular compliance, data in double AQP-null mice surprisingly indicate that these channels could also influence ventricular distensibility. However, the specific mechanism behind this remains unclear and warrants further experimental exploration. Questions to address include whether this alteration is related to reduced ventricular volume, changes in the biophysical properties of cerebral parenchyma, or systemic changes that secondarily affect cerebral compliance.

In addition to AQP1 and AQP4, another channel, Aquaglyceroporin 9 (AQP9), is present in the brain. It is found in astrocytes at the glial limitans, endothelial cells in pial vessels, and specific neuronal groups [79,80]. The role of AQP9 in CSF dynamics and its impact on cerebral pathologies is a burgeoning area of research that needs further experimental and clinical investigation [81,82].

Importantly, most knowledge about the role of AQP1 and AQP4 in CSF homeostasis comes from animal models. However, growing evidence suggests that these aquaporins serve similar functions in humans and could be implicated in various pathophysiological conditions [83,84]. Additionally, a newly identified aquaporin, AQP11, has been reported to be expressed in the brain [85,86]. Future work should consider whether alterations observed in double AQP-null mice might be due to compensatory mechanisms involving other aquaporins or changes in brain water content.

## 5. The Glymphatic System: An Essential Framework for Brain Health

### 5.1. Comprehensive Breakdown of the Glymphatic System’s Architecture and Its Functional Significance

The glymphatic system, a groundbreaking discovery in neuroscience, serves as a macroscopic waste clearance pathway that employs a specialized network of perivascular channels created by astroglial cells. This system is essential for the efficient removal of soluble proteins and metabolites from the central nervous system. Beyond its role in waste clearance, the glymphatic system may also assist in the distribution of vital substances such as glucose, lipids, amino acids, and neurotransmitters across the brain. Notably, the system is primarily active during sleep, suggesting that the biological necessity for sleep may stem from the brain’s need to eliminate potentially harmful waste products, including β-amyloid [87].

Recent research indicates that cerebrospinal fluid (CSF) and interstitial fluid (ISF) engage in a continuous exchange, facilitated by convective influx along the periarterial space [71]. Driven by factors such as arterial pulsatility, respiratory rhythms, and pressure gradients, CSF flows from the subarachnoid space into the Virchow-Robin spaces. The perivascular space acts as a low-resistance pathway for CSF influx. The migration of CSF into the intricate brain parenchyma is enabled by Aquaporin 4 (AQP4) water channels, which are highly polarized and localized in the astrocytic endfeet that envelop the brain vasculature [71,88].

This influx of CSF into the brain parenchyma triggers convective fluxes of ISF within the tissue, leading it toward perivenous spaces adjacent to large, deep veins. Eventually, the ISF is gathered in these perivenous spaces and drains towards the cervical lymphatic system [89,90]. The system of convective fluid fluxes, characterized by its rapid exchange between CSF and ISF, has been dubbed the glymphatic system. This name reflects its functional similarities to the peripheral lymphatic system and emphasizes the critical role of glial AQP4 channels in facilitating fluid transport.

In 2012, the inner workings of the glymphatic system were detailed for the first time through in vivo studies using two-photon microscopy in mice [71]. Iliff and colleagues marked cerebrospinal fluid (CSF) with fluorescent tracers injected into the cisterna magna, which allowed them to observe that the CSF rapidly enters the brain along cortical pial arteries. This influx was followed by its movement into the Virchow-Robin spaces along penetrating arterioles. Importantly, it was found that the tracers in the CSF were not diffused randomly throughout the brain tissue. Instead, they entered via a specific periarterial pathway that is closely aligned with vascular smooth muscle cells and bounded by the perivascular astrocytic endfeet. Further ex vivo studies showed that these tracers exited the brain mainly along central deep veins and lateral-ventral caudal rhinal veins [71].

Subsequent analyses revealed that this directed movement of CSF via periarterial pathways into the brain parenchyma assists in clearing out interstitial solutes, directing them towards perivenous drainage channels. This finding has significant implications for neurodegenerative diseases, such as Alzheimer’s, which are characterized by protein accumulations, including amyloid plaques and tau tangles [91,92].

To investigate the role of the glymphatic system in clearing β-amyloid, a hallmark of Alzheimer’s disease, Iliff et al. injected fluorescent or radiolabeled amyloid β1–40 into the striatum of mice. Their observations confirmed that β-amyloid was rapidly eliminated from the brain via the glymphatic system’s paravenous efflux pathways [71].

Moreover, imaging studies in mice lacking Aquaporin 4 (AQP4) showed significant disruptions in the glymphatic system. Specifically, there was approximately a 65% reduction in the flux of CSF through the brain tissue when compared to wild-type control mice. Additionally, the clearance rate of intrastriatally-injected radiolabeled β-amyloid was reduced by 55% [71]. These findings led to the proposition that the AQP4-mediated glymphatic pathway serves as a crucial mechanism for the removal of interstitial fluid solutes, including waste products, from the brain’s parenchyma [88].

### 5.2. AQP4-Centric Discussion on the Glymphatic Pathway, Detailing How Aquaporin Malfunctions Might Hinder the System

#### Convective CSF Fluxes in Aging and Pathology

Glymphatic activity decreases sharply during aging-Recent research has shown a striking decrease in glymphatic function in older mice compared to their younger counterparts, with an estimated 80–90% reduction in activity [93]. This decline encompasses both the influx of CSF tracers into the brain and the clearance of radiolabeled β-amyloid and inulin. It has been suggested that reactive gliosis, characterized by the hypertrophy of GFAP+ astrocyte processes, may be a contributing factor to this age-related decline in glymphatic function. However, the exact mechanisms linking changes in GFAP expression to diminished glymphatic activity remain unclear [94].

In younger animals, Aquaporin 4 is localized to the astrocytic endfeet and plays a pivotal role in facilitating the exchange of cerebrospinal and interstitial fluid along periarterial influx pathways (Figure 2), as well as aiding in the clearance of interstitial solutes through perivascular drainage paths. Previous research has shown that the genetic deletion of AQP4 leads to a roughly 65% impairment in CSF-ISF exchange and a 55% reduction in the clearance of β-amyloid [95]. In aging brains, there is a partial loss of this vascular polarization of astrocytic AQP4 (Figure 2). Specifically, AQP4 is no longer solely confined to the astrocytic endfeet but is also found in the parenchymal processes of astrocytes [96].

This loss of AQP4 polarization in older brains suggests that the age-related decline in glymphatic function could be partially attributed to dysregulation in astroglial water transport. Other factors that might contribute to the decrease in glymphatic activity with age include a 66% reduction in CSF production and a 27% decrease in CSF pressure [97]. Additionally, the arterial walls become stiffer with age, leading to a decrease in arterial pulsatility, which is one of the key drivers of glymphatic influx [98].

The observed decline in glymphatic function due to aging is especially significant given that age is the most significant risk factor for neurodegenerative diseases. A compromised glymphatic system in older individuals could potentially lead to the accumulation of misfolded and hyperphosphorylated proteins, thereby making the brain more susceptible to neurodegenerative diseases or even accelerating the progression of cognitive decline.

### 5.3. Correlation between Compromised Glymphatic Functionality and Neurodegenerative Disorders

Traumatic brain injuries (TBIs), commonly observed in military personnel and athletes, are associated with an elevated risk of premature dementia and Alzheimer’s disease [99]. Numerous studies have indicated that both recurring traumatic incidents, as well as single events of moderate to severe head trauma, can result in ongoing neurodegeneration. The mechanism by which only certain individuals develop chronic traumatic encephalopathy after experiencing a similar degree of initial brain injury remains elusive [100]. TBIs result in the release of β-amyloid peptide and C-tau, a proteolytic derivative of MAP-tau, which is a prominent intracellular microtubule protein in axons. Notably, C-tau, due to its vast release correlating with TBI severity, serves as a brain injury biomarker [101]. A prevailing theory posits that significant surges in interstitial tau can lead to its cellular intake, initiating fibrillary aggregates. These aggregates attract more tau, fostering the formation of neurofibrillary tangles and facilitating a prion-like progression of the disease. Moreover, TBIs are connected to the development of extensive astroglial scars and the sustained activation of inherent neuroinflammation. In a study focusing on repetitive moderate TBIs, the influx of cerebrospinal fluid (CSF) into the brain was compromised in the ipsilateral hemisphere from the first day post-injury, and this reduced glymphatic functionality endured for at least 28 days post-injury. This significant decline in glymphatic activity was linked to glial scars marked by enlarged GFAP-positive processes in the ipsilateral hemispheres. Moreover, an AQP4 mislocalization from the vascular endfeet to parenchymal processes was observed, paralleling the AQP4 misplacement seen in aging processes [93]. Through intracortical injections of human tau, Iliff et al. managed to monitor the tau clearance pathway, noting that human tau congregated around large veins [97]. The residual tau in the tissue was in line with diminished glymphatic clearance, underscoring the pivotal role of CSF-mediated tau removal through glymphatic routes in mitigating subsequent neuronal damage post-TBI. Another investigation employing magnetic resonance imaging (MRI) to evaluate glymphatic function revealed that head traumas, such as subarachnoid hemorrhages, considerably weaken glymphatic function [102]. Introducing freshly isolated arterial blood into the CSF inhibited the CSF’s influx pathways into the brain, excluding the cerebellum, indicating that cerebral hemorrhages can extensively suppress glymphatic functionality. In this subarachnoid hemorrhage model, the use of a tissue-type plasminogen activator, which eliminates fibrin clots, enhanced glymphatic perfusion. An embolic ischemic stroke temporarily hindered glymphatic flow shortly post-ischemia; however, functionality was naturally restored 24 h post-transient ischemia [102], suggesting that brief glymphatic function interruptions, resulting from reduced arterial pulsation or mild stroke-induced perivascular pathway blockages, can be reversible and contribute to enhanced recuperation.

## 6. Implications of Aquaporins in Degenerative and Acute Brain Pathologies

### 6.1. Profiling Each Degenerative Disease (iNPH, PD, AD) and Its Associated Aquaporin Dysregulations

#### 6.1.1. Parkinson’s Disease

Neurodegenerative diseases impact millions globally, constituting a diverse set of disorders that specifically target certain regions of the Central Nervous System (CNS). These diseases typically result in a sustained decline in cognitive or motor functions, contingent upon the distinct type of neuronal cells undergoing selective degeneration [103]. Predominantly, these pathological states are correlated with aging. Indeed, as life expectancy has risen over recent decades, the prevalence of these age-related disorders has correspondingly increased [104]. Neuronal harm and oxidative stress, which are fundamental events in the onset of these conditions, precipitate an upsurge in the production of pro-apoptotic and pro-inflammatory cytokines. Concurrently, there’s a disruption in the balance of water, extracellular ions, and amino acid neurotransmitters. Given the perturbed brain homeostasis observed in many of these diseases, some researchers have postulated a potential involvement of the Aquaporin (AQP) protein family in the pathogenic mechanisms underlying these conditions [105].

#### 6.1.2. Dopamine Regulation of AQP4 Expression

In the realm of neurodegenerative diseases, a pivotal role is played by neural stem cells. Throughout adulthood, these cells have the capability to proliferate and differentiate into either new neurons or glial cells [106]. Intriguingly, studies have highlighted that adult neural stem cells display glial-associated properties in both in vivo and in vitro environments. As a testament to this, these stem cells express GFAP, a protein considered a hallmark for fully differentiated astrocytes [107]. Notably, alterations in the count of GFAP-expressing cells have been implicated in neurodegenerative disorders, including Parkinson’s disease [108].

Dopamine (DA), a central neurotransmitter, has been found to invigorate the proliferation of progenitor cells. This effect is observed not just in the striatum but also in the subventricular zone of mature brains. Building on this, a recent investigation led by Kueppers et al. postulates that DA orchestrates the proliferation of striatal astrocytes in culture through the mediation of Aquaporin-4 [109]. Their results delineate a scenario wherein DA prompts a reduction in AQP4 expression in striatal glial cells when studied in vitro.

However, the narrative surrounding AQP4’s role in cellular proliferation is replete with ambiguities. Saadoun et al. observed a stable rate of proliferation in astrocytes cultured from AQP4-deficient transgenic mice. Contrasting this, Nicchia et al. identified a pronounced ~70% decline in cultured astrocyte numbers after the cells were subjected to short interference RNA (siRNA) treatments targeting AQP4 [110,111]. This underscores the necessity for in-depth in vivo lesion studies to validate these findings. Furthermore, understanding AQP4 expression in a damaged striatum is imperative, especially given observations of elevated AQP4 mRNA levels in the substantia nigra following a 6-hydroxydopamine (6-OH-DA) lesion [112].

These findings coalesce to underline the hypothesis that DA can modulate astrocyte proliferation. This strengthens the assertion that neurodegenerative diseases, where dopaminergic transmission is disrupted (such as PD), are inextricably linked with alterations in astrocyte proliferation. Conclusively, these insights open up the potential therapeutic avenue of modulating AQP4 as an intervention strategy in treating PD.

#### 6.1.3. Mitochondrial AQP9 in PD Brains

Within the scope of neurodegenerative disorders, a potentially significant yet conjectural association has been proposed between Aquaporin-9 (AQP9) and Parkinson’s disease [113]. Within the cerebral environment, AQP9—a channel facilitating the movement of water and certain solutes—is manifested in astrocytes, brain stem catecholaminergic neurons, as well as specific subgroups of midbrain dopaminergic and hypothalamic neurons. An intriguing observation is the pronounced presence of AQP9 within the mitochondrial inner membranes, hinting at its potential role in supporting neuronal metabolism. Especially compelling is the notion that aberrations in mitochondrial AQP9 within dopaminergic neurons might be linked to the heightened susceptibility of these neurons to PD [113].

Given the presumptive significance of mitochondrial AQP9, Yang et al. embarked on an exhaustive exploration to discern the putative functional repercussions of its expression [114]. Their investigations centered on determining the transport functionality of AQP9 within mitochondrial inner membranes. Through empirical assessments, they gauged the permeabilities of water and glycerol in brain mitochondria, juxtaposing these with measurements derived from tissues devoid of AQP9 expression [115]. Surprisingly, their findings indicated no discernible variances in water or glycerol permeabilities across different mitochondrial sources. These findings, in essence, cast doubts on the postulated role of aquaporins within the mitochondrial environment. Despite these insights, whether AQP9’s expression and activity can be harnessed as therapeutic avenues to augment PD treatment remains an open-ended inquiry (Figure 3).

#### 6.1.4. Idiopathic Normal Pressure Hydrocephalus (iNPH)

Idiopathic normal pressure hydrocephalus (iNPH) is recognized as a distinct form of dementia, wherein intervention through cerebrospinal fluid diversion can yield favorable outcomes. Pioneering studies leveraging magnetic resonance imaging with CSF tracers have highlighted compromised CSF tracer clearance from specific cerebral regions, notably the entorhinal cortex, in iNPH patients. This compromised clearance mechanism, especially for waste solutes such as soluble amyloid-β, might be central to the neurodegenerative processes and cognitive decline characterizing iNPH.

The primary objective of the study at hand was to investigate potential alterations in the subcellular localization of aquaporin-4 water channels in relation to iNPH. Notably, AQP4 is known to play a pivotal role in modulating CSF flow and facilitating the glymphatic removal of brain metabolites. To accomplish this, cortical brain biopsy samples from 30 iNPH patients and 12 control subjects were scrutinized using AQP4 immunogold cytochemistry.

Utilizing electron microscopy, the study unveiled a markedly diminished presence of AQP4 water channels within the astrocytic endfoot membranes juxtaposed to cortical microvessels in iNPH patients, relative to the control group. Further analysis divulged a statistically significant association between AQP4 densities oriented perivascularly and those directed towards the parenchyma. However, the decline in AQP4 densities oriented towards the parenchyma, while observed, was not statistically significant in iNPH cases.

The findings suggest that iNPH is associated with a suppressed perivascular expression of AQP4. Such diminished AQP4 presence might adversely impact the glymphatic circulation, hindering efficient waste removal and potentially fostering neurodegeneration. Therefore, strategies aimed at reinstating the optimal perivascular distribution of AQP4 could be postulated as innovative therapeutic avenues for addressing iNPH [116].

### 6.2. Insights into Aquaporin Behavior during Acute Cerebral Events, Such as Stroke or Traumatic Injuries

Aquaporins, water channel proteins found in the brain, have attracted substantial research attention given their potential implications in both physiological and pathological contexts [117]. Among these, AQP4 has been the focal point of numerous studies, especially in relation to various brain conditions, spanning acute afflictions such as stroke and traumatic brain injury to chronic autoimmune neurodegenerative ailments. As of now, there are no targeted therapeutic interventions specifically designed to modulate the water transport activity of these channels. However, accumulating experimental data underscores the pivotal nature of AQPs, suggesting their profound relevance in future research endeavors. For instance, curtailing water channel activity early on might ameliorate edema development in instances of brain injuries, while during the later stages of certain diseases, AQPs play a crucial role in facilitating water clearance from the brain to blood vessels [118].

A predominant characteristic of many brain diseases, ranging from stroke, traumatic brain injuries, and brain tumors to inflammation, is edema. This phenomenon denotes the accumulation of water due to imbalances in brain osmotic homeostasis. A primary ramification of edema is brain swelling, which can exacerbate secondary complications, such as compromised brain perfusion. Despite its long-standing recognition in both clinical and pre-clinical realms, the molecular and cellular intricacies underpinning edema genesis and resolution remain relatively elusive. Furthermore, current therapeutic interventions fall short of effectively curbing edema onset or progression in various brain conditions. Thus, the identification of brain AQPs provided a glimmer of optimism in the quest for novel therapeutic approaches to counteract edema. The insights gleaned over the past decade and a half regarding the potential of AQPs as therapeutic targets for edema will be delineated, followed by a brief overview of the initial phases of edema formation [117].

#### 6.2.1. Edema Build-Up Phase: Anoxic, Ionic and Vasogenic Edema

Cerebral edema, for the past four decades, has been predominantly categorized into two principal types: cytotoxic and vasogenic [119]. Conventionally, cytotoxic edema denotes the accumulation of intracellular water without any disruption of the blood–brain barrier (BBB). In contrast, vasogenic edema emerges post-BBB disruption, instigating protein diffusion from the bloodstream into the tissue, subsequently leading to water buildup in the extracellular matrix. This longstanding bifurcation, however, is now considered an oversimplification, particularly in light of recent insights into the molecular shifts during edema onset and the evolving understanding of BBB properties. Instead, with respect to vascular brain injuries, edema’s initiation phase can be more accurately segmented into three primary categories: anoxic, ionic, and vasogenic edema [120].

Anoxic edema is typified by the immediate swelling of astrocytes and neuronal dendrites ensuing moments after a deprivation of oxygen and glucose, particularly in cerebrovascular ailments. This deficiency in essential nutrients triggers significant perturbations in cellular ionic gradients due to the non-operational energy-dependent co-transporters. Consequently, this facilitates an extensive ion influx into cells, a manifestation of which is a gradual surge in the extracellular K+ levels, leading to subsequent water entry and resultant swelling initially in astrocytes and later in neuronal dendrites [121]. Rapidly following this, anoxic edema transitions into ionic edema. The depletion of vital nutrients similarly impacts the ionic gradients in endothelial cells, manifesting in changes such as transcapillary sodium flux leading to tissue edema. This endothelial distress results in an initial transient BBB leakage observed in conditions such as stroke and traumatic brain injuries (TBI) [122]. Such changes culminate in enhanced water influx through endothelial cells, leading to brain swelling, as exemplified in stroke models observed within the first 30 min of reperfusion, coupled with further intensified BBB permeability [123]. The ensuing vasogenic edema is characterized by heightened permeability to plasma proteins such as albumin, stemming from a comprehensive disruption of endothelial tight junctions, extracellular matrix degradation, and potential augmentation in transendothelial cell transport via the transcytosis mechanism [119].

However, it is pivotal to highlight that such delineation primarily pertains to brain injuries characterized by acute cerebrovascular anomalies, and might not be wholly applicable to other neurological conditions, such as brain tumors [124]. This underscores the importance of recognizing that clinical interventions focusing exclusively on osmotic challenges might not effectively treat cerebral edema. The rationale behind this is the intricate and varied molecular mechanisms steering edema development. While the precise functionalities of cerebral AQPs in this context remain a topic of active research, their strategic localization and inherent nature as water channel proteins suggest their significant implications in cerebral edema dynamics. The subsequent sections delve into the roles of AQPs in the context of this refined classification of edema formation.

#### 6.2.2. Contribution of AQPs in Edema Formation and Resolution

Aquaporins, specifically AQP1, 4, and 9, have demonstrated alterations in their expression levels in various brain disorders, as seen in both rodent models and human specimens [125]. Of these, AQP4 has been the primary focus of many investigations, as its expression patterns appear to mirror the progression of edema in numerous neurological conditions [126]. A significant advancement in the exploration of AQP4’s role in edema came with the creation of AQP4 knockout mice (AQP4−/−) by Dr. Verkman’s team [127].

Interestingly, these AQP4−/− mice did not show any substantial structural or physiological deviations. However, a notable observation was the expansion of their extracellular space by roughly 20% in comparison to their wild-type (WT) counterparts [128]. The outcomes presented by Dr. Verkman’s group generated a hypothesis suggesting a dual functionality for AQP4 in the context of edema: AQP4 could exacerbate edema during its formation phase, but conversely, it might play a beneficial role during the edema resolution phase. Yet, the lack of specific pharmaceutical agents capable of acutely and selectively inhibiting AQP4 made it challenging to validate this dual-role theory post-brain injury. It was only recently that this hypothesis was put to the test in vivo, utilizing a siRNA methodology.

#### 6.2.3. AQP4 and Edema Build-Up

The role of Aquaporin-4 in cerebral edema is complex, exhibiting a spectrum of expression profiles across various pathological conditions including traumatic brain injury, ischemic stroke, and subarachnoid hemorrhage. These conditions each display unique fluctuations in AQP4 expression levels [124,126,129]. Specifically, in ischemic models involving transient middle cerebral artery occlusion, AQP4 is acutely up-regulated in the astrocyte endfeet adjoining blood vessels, reaching peak levels approximately 1 h post-stroke onset. This elevation in AQP4 expression is spatially and temporally correlated with the extent of cerebral edema and is most prominent in the peri-infarct region as well as the prospective lesion site [123,126].

This initial surge in AQP4 is implicated in the genesis of ionic cerebral edema and astrocyte swelling. Intriguingly, in more severe ischemic conditions, this up-regulation is not observed, leading to the hypothesis that under extensive tissue stress, the brain may lack the capacity for rapid AQP4 synthesis during the early reperfusion phase. Furthermore, the ischemic conditions induce a shift in the ratio of AQP4 isoforms, AQP4-m1 and AQP4-m23, suggesting a potential disruption of orthogonal arrays of particles (OAPs). The functional implications of these alterations, however, remain to be elucidated [130].

The modulation of AQP4 expression is influenced by a multitude of factors, including injury severity, model system, and age, further complicating the interpretation of its role in edema formation [129]. Genetic studies utilizing AQP4 knockout mice (AQP4−/−) have also yielded conflicting outcomes. For example, AQP4 deletion conferred protective effects in spinal cord injury models, characterized by reduced edema and lesion size in the acute phase [131]. Contrarily, another study observed improved functional outcomes in wild-type mice relative to AQP4−/− mice in long-term follow-up post-spinal cord injury [132]. Such discrepancies point to the limitations inherent in employing AQP4−/− mice as a tool to ascertain AQP4’s pathophysiological roles.

In recent advancements, small interfering RNA (siRNA) targeting AQP4 (siAQP4) has been developed to transiently down-regulate AQP4 expression and has been shown to reduce water mobility [133,134]. In juvenile traumatic brain injury models, this targeted molecular intervention was associated with reduced edema and cognitive improvement at two months post-injury, suggesting siAQP4 as a potential therapeutic modality [134]. Despite these developments, the role of AQP4 in edema formation remains complex and not fully understood. Overall, findings indicate that AQP4 may have a dual function, either exacerbating or mitigating edema, depending on a variety of factors, including the specific pathological context.

#### 6.2.4. Edema Resolution in Acute Brain Disease: Role of AQP in Water Clearance

The hypothesis that Aquaporin-4 has a dual role in cerebral edema—being deleterious during edema formation while beneficial during its resolution—has gained empirical support despite the absence of conclusive evidence. Initial evidence for this comes from experiments using AQP4 knockout mice (AQP4−/−), which revealed that intracranial pressure increased significantly when a saline solution was infused into the brain parenchyma compared to wild-type mice [8]. Furthermore, multiple studies utilizing magnetic resonance imaging (MRI) have demonstrated that elevated AQP4 expression is temporally correlated with the resolution of edema in various pathological states, including stroke, traumatic brain injury (TBI), and neuroinflammatory lesions [119,133,135].

Specifically, AQP4 expression typically escalates 48 h post-insult in models of stroke, TBI, and neuroinflammatory lesions. This augmented expression is frequently localized to the astrocyte endfeet adjacent to blood vessels, as well as the astrocyte processes and glia limitans [126,129]. Such spatial distribution of heightened AQP4 expression suggests its potential role in facilitating the clearance of edematous fluid via the subarachnoid space. For instance, in juvenile traumatic brain injury (jTBI) models, elevated levels of AQP4 in the glia limitans appear to counterbalance water accumulation at one- and three-days post-injury (as indicated by higher T2 values), leading to a normalization of both AQP4 and T2 levels by day seven [129].

In neuroinflammatory lesion models in rats, apparent diffusion coefficient (ADC) time course studies revealed a bifurcation in AQP4 expression patterns, distinguishing its minor elevation during the edema formation phase from its pronounced increase during the edema resolution phase. Importantly, peak AQP4 expression levels coincided with significantly elevated ADC values, providing further support for its role in edema resolution.

The role of other aquaporin family members, such as Aquaporin-9, in the resolution of cerebral edema warrants exploration. While AQP9 is up-regulated in reactive astrocytes seven days post-ischemic injury along the infarct border, its expression pattern does not show a strong correlation with the extent of cerebral swelling, in contrast to AQP4 [136]. Moreover, this alteration in AQP9 expression is not universally observed across various models of brain pathology. In gerbil models of ischemic stroke, AQP9 is expressed in hippocampal pyramidal neurons in regions CA1, CA2, and CA3 as early as 6 h post-stroke onset, a pattern distinct from other stroke models. Taken together, these observations suggest that AQP9 may have a limited role in edema regulation following brain injury. Notably, pyramidal neurons typically do not express this channel under physiological conditions, suggesting that its expression may be induced by metabolic stressors. Despite this, the functional implications of increased AQP9 expression in neurons and astrocytes remain unclear. A possible role for AQP9 in astrocyte energy metabolism has been suggested, indicating that enhanced AQP9 expression could facilitate glycerol utilization as an alternative metabolic substrate, thereby potentially aiding in neuronal recovery [126].

In contrast to AQP9, AQP4 appears to collaborate with other astrocytic proteins such as Connexin-43 (Cx43) and the potassium channel Kir4.1 in managing cerebral edema. Cx43 forms gap junctions between astrocytes and plays a critical role in solute and water diffusion within the astrocytic network. Intriguingly, Cx43 expression is down-regulated when AQP4 is silenced using RNA interference in primary astrocyte cultures, which could potentially impair astrocytic connectivity. Furthermore, Kir4.1 is co-localized with AQP4 in the astrocyte endfeet and has been implicated in potassium reuptake impairment in AQP4 knockout mice in epilepsy models. Notably, potassium flux has been associated with water movement during astrocyte swelling in spinal cord injury models [137,138,139]. Thus, it is plausible that AQP4, Cx43, and Kir4.1, all expressed in astrocytes, may function synergistically in the regulation and resolution of cerebral edema following brain injury.

#### 6.2.5. Chronic Changes of Brain AQP: Relation with Water Homeostasis Dysfunction?

Aquaporins, traditionally implicated in water homeostasis both in physiological and pathological contexts, have been increasingly associated with a broader range of cellular functions, including cell migration and gas diffusion [140]. Specific isoforms, such as AQP1, AQP4, and AQP9, manifest elevated expression in both brain tumors and peritumoral tissues. The augmented presence of these AQPs in astrocytes within peritumoral regions may be related to edema formation, potentially due to altered tissue homeostasis and elevated metabolic rates. Additionally, these AQPs may facilitate gas diffusion, playing a role in the clearance of excess CO_2_ and the diffusion of O_2_. In rodent models of spinal cord injury (SCI), an upregulation of AQP1 is observed in astrocytes as well as neurons. AQP1 expression has also been documented in neuronal processes in the dorsolateral septum at various time points following juvenile traumatic brain injury (jTBI) [129]. While astrocytic expression of AQP1 may be associated with cerebrospinal fluid secretion during cyst formation, its co-localization with growth-associated protein-43 (GAP-43) in neurons suggests a role in neuroplasticity and repair after injury. Furthermore, AQP1 knockout mice exhibit reduced pain responses, highlighting its potential relevance to the cognitive pathways associated with pain sensation, although a consensus regarding AQP1’s role in pain processing has yet to be established [141].

Longitudinal changes in AQP4 expression have also been noted; its expression remains elevated up to 28 days in the perilesional area following stroke and SCI [142]. AQP4 has been implicated in astrocyte migration, particularly in the formation of the glial scar, by facilitating the water influx necessary for filopodia formation and promoting cellular adhesion among astrocytes [143]. This heightened expression of AQP4 in astrocytes is likely to contribute to glial scar formation, aiding in the establishment of a new tissue barrier at the borders of post-injury cavities.

In the context of chronic neurodegenerative diseases such as Alzheimer’s disease, as well as post-jTBI, a reduction in AQP4 expression has been observed around blood vessels in association with beta-amyloid deposition. This decrease in AQP4 levels could potentially disturb water homeostasis within the neurovascular unit, subsequently attenuating perivascular flow and impeding the effective clearance of toxic substances, such as beta-amyloid, from the brain tissue [71].

### 6.3. The Cascading Effects of Altered AQP Expression in Autoimmune Conditions, with a Focus on NMO

Aquaporins are not just pivotal in regulating water homeostasis; they also have roles in the immune system, specifically within both the innate and adaptive arms. In human blood leukocytes, AQP1 and AQP9 are expressed and show upregulation upon stimulation with lipopolysaccharide (LPS) either intravenously or in vitro [144]. AQP9 expression is also elevated in activated polymorphonuclear leukocytes in patients suffering from systemic inflammatory response syndrome (SIRS) and infective endocarditis [145].

B and T lymphocytes, key players in adaptive immunity, have been found to express AQP1, AQP3, and AQP5. Similarly, immature dendritic cells (DCs), which are integral to the innate immune system, express AQP3 and AQP5. In these immune cells, AQP expression is often correlated with their activation and proliferation. Notably, AQP9 is the most highly expressed isoform in DCs and shows further upregulation upon LPS stimulation. In a mouse model of induced colitis with AQP9 knockout (AQP9-KO), the absence of AQP9 did not offer complete protection against colitis-associated inflammation but did diminish the DC-mediated inflammatory response [146].

Human primary blood-derived macrophages and neutrophils, critical components of the innate immune system, display high levels of AQP9, which also sees upregulation at both transcript and protein levels when stimulated with LPS [147]. AQP3, another isoform, is also sensitive to LPS stimulation in monocytic THP-1 cells, a model often used to study inflammation. The inhibition or silencing of AQP3 in these cells leads to partial blockage of LPS priming and a reduction in the production of key inflammatory cytokines such as interleukin-6 (IL-6), pro-IL-1β, and tumor necrosis factor-alpha (TNF-α). This implicates a functional connection between AQP3 and Toll-like receptor 4 (TLR4) during the priming of macrophages [147].

Moreover, a separate study on THP-1 cells revealed an increase in AQP1 expression after LPS treatment, while AQP5 mRNA levels decreased [148]. Elevated AQP9 expression was also found in neutrophils in SIRS patients compared to healthy controls [149].

AQPs appear to play multifaceted roles in the immune response, particularly during inflammation. They are not only expressed in various immune cells but also undergo regulation in response to inflammatory stimuli, as summarized in Table 1. This highlights their potential significance in understanding and possibly treating inflammatory conditions.

Neuromyelitis optica spectrum disorders (NMOSD) are a range of inflammatory demyelinating diseases (IDDs) that primarily affect the optic nerves and spinal cord but can also extend to the brain and, in rare instances, muscles. Brain lesions in NMOSD often localize to areas with high AQP4 expression, such as the circumventricular organs (responsible for intractable nausea and vomiting) and the diencephalon (linked to sleep disorders, endocrine imbalances, and the syndrome of inappropriate antidiuresis). Up to 10% of NMOSD patients even fulfill the Barkoff criteria for multiple sclerosis when evaluated through MRI [153].

One of the hallmarks of NMOSD is the presence of autoantibodies against AQP4, known as AQP4-IgG or NMO-IgG, detected in 60–90% of NMO patients [154,155]. AQP4 is not only found on the astrocytes in the central nervous system (CNS) but also in skeletal muscle and various epithelial cells such as those in the kidney, stomach, and exocrine glands. AQP4-IgG was initially perceived as a mere marker of the disease, possibly related to astrocyte damage. However, mounting evidence now suggests that AQP4-IgG plays a pathogenic role in NMO.

When AQP4-IgG binds to AQP4 on astrocytes, this initiates a cascade of immunological responses, primarily through complement-dependent cytotoxicity. This leads to the invasion of leukocytes into the CNS, the release of cytokines, and the disruption of the blood–brain barrier. These series of events are thought to culminate in the death of oligodendrocytes (cells responsible for myelination), resulting in myelin loss and, ultimately, neuronal death. This cascade explains the neurological deficits observed in patients with NMO.

Given the rapidly evolving knowledge on the immunobiology of AQP4 autoimmunity, there is a growing need for ongoing revisions in the diagnostic criteria for NMOSD [153]. As our understanding broadens, the role of highly specific assays that can detect pathogenic AQP4-IgG targeting the extracellular domains of AQP4 becomes increasingly crucial.

The size of AQP4-IgG (autoantibodies against AQP4) presents a steric challenge when it comes to binding with AQP4, which is a tetramer consisting of four separate monomers and, by extension, four distinct water pores. This size mismatch suggests that it would be unlikely for a single AQP4 tetramer to bind with more than one AQP4-IgG molecule. Therefore, significant inhibition of water permeability by AQP4-IgG appears to be theoretically implausible.

Backing this notion, despite extensive research efforts, no small-molecule inhibitors of AQP4 have been identified to date. Several cell culture studies have concluded that AQP4-IgG does not inhibit the water permeability of AQP4 [156,157]. In experiments that used a stopped-flow light scattering method to evaluate AQP4 water permeability, neither high concentrations of serum from multiple NMO patients nor monoclonal NMO antibodies were found to significantly affect AQP4 function [158]. This approach, involving plasma membrane vesicles from AQP4-expressing cells and AQP4-reconstituted proteoliposomes, is sensitive enough to detect variations in water permeability less than 5% and is not influenced by factors such as internalization, unstirred layers, or ion/solute transport.

However, it is worth mentioning that there is one conflicting study that utilized a time-to-lysis assay on Xenopus oocytes and claimed that AQP4-IgG inhibits AQP4 water permeability. This method is considered an inaccurate surrogate for measuring osmotic water permeability [159].

Except for this one conflicting study by Hinson et al. [159], the prevailing evidence supports the idea that AQP4-IgG does not inhibit the water permeability of AQP4. Given this, the role of AQP4-IgG in the pathology of Neuromyelitis Optica Spectrum Disorders may lie elsewhere, possibly in initiating immunological cascades that lead to tissue damage rather than directly affecting water transport through AQP4.

## 7. Aquaporins at the Intersection of Oncology and Neurology

### 7.1. Unraveling the Possible Links between Aquaporin-Mediated Processes and Brain Tumorigenesis

AQP1, primarily known for its role in water transport, has additional functions that are intriguing both scientifically and medically. One of these is its capability to act as a cyclic nucleotide-gated cation channel activated mainly by cGMP and, to a lesser extent, by cAMP [160]. Research by Yu and colleagues suggests that the interaction of cGMP with an arginine-rich cytoplasmic Loop D in AQP1 leads to a conformational change that may mediate the gating of its central ion channel [161,162].

Interestingly, AQP1 overexpression has been documented in a variety of human cancers, including those of the biliary duct, bladder, brain, breast, cervix, colon, lung, nasopharynx, and prostate [163,164]. Specifically in the case of colon cancer, Moon and colleagues have shown that AQP1 is expressed in colonic adenoma and both primary and secondary colon cancer, but is absent in normal colonic mucosa [164]. This suggests that AQP1 may play a role in the early stages of colon cancer development. Additionally, the expression levels of AQP1 have been found to correlate with key clinical prognostic indicators such as histological grade, lympho-vascular invasion, and nodal involvement [163,165].

The link between AQP1 and cancer progression has garnered significant attention, leading to numerous reviews on the subject. These reviews often focus on the potential of AQP inhibitors as therapeutic agents in cancer treatment [166,167]. Given the multifaceted roles of AQP1, from water transport to ion channel gating and its association with various types of cancer, the protein appears to be a critical player in both physiology and pathophysiology, including tumorigenesis. As such, understanding its function and regulation could offer valuable insights into the development of novel therapeutic approaches for a range of diseases, including cancer (Figure 4).

### 7.2. Implications of Aquaporins in Neoplastic Cell Migration, Invasiveness, and Angiogenesis

#### 7.2.1. AQP1-Modulated Tumor Cell Migration and Invasion

The role of AQP1 in cancer progression extends beyond its well-known function in water transport, particularly implicating it in the critical processes of tumor cell migration and invasion. Hu and Verkman found that AQP1 accelerates the migration of specific mouse melanoma and breast cancer cell lines in vitro. Notably, they observed polarized AQP1 expression at the leading edge of migrating cells. In vivo studies further revealed that AQP1 promoted cancer cell extravasation and lung metastases [168]. Jiang’s work supports these findings, showing that altering AQP1 expression levels directly influences the migratory behavior of HT20 human colon cancer cells both in vitro and in vivo [169].

One proposed mechanism by which AQP1 facilitates tumor cell migration involves osmotic water flow across the plasma membrane. This flow is thought to be induced by an osmotic gradient created by actin depolymerization and active solute influx at the cell’s leading edge [170]. Water influx via AQP1 then increases hydrostatic pressure, leading to local expansion of the plasma membrane and actin re-polymerization to stabilize cell membrane protrusions [6]. Chen and colleagues also provided evidence that AQP1’s role extends to facilitating intracellular parasite invasion through mechanisms involving localized water influx [171].

An alternative explanation focuses on AQP1’s role in changing the cell shape and volume as tumor cells navigate through confined spaces. The water flow facilitated by AQP1 helps generate the hydrostatic forces needed for this process. Actin polymerization and depolymerization as well as ionic fluxes across the membrane could support this osmotic water flow [124].

Adding a more nuanced layer to these mechanisms, Stroka and colleagues proposed an “Osmotic Engine Model”, where cell migration in confined spaces is driven not just by actin and myosin interactions but also through water permeation and ion transport mediated by AQPs and Na+/H+ pumps [172].

The implications of these findings are profound, as the acquisition of migratory and invasive capabilities is a key step in cancer metastasis, which accounts for the majority of cancer-related deaths. Understanding AQP1’s multifaceted role in these processes could thus offer crucial insights for the development of new therapeutic strategies targeting cancer metastasis.

The role of ion channels and transporters in cellular migration is increasingly recognized as critical. These components regulate various cellular mechanisms, such as intracellular calcium levels (Ca2+i), pH balance both inside and outside the cell, cellular membrane potential, and volume [173]. During directional cell migration, specific ion channels and transporters, including AQPs, K^+^ channels, and Na^+^/H^+^ exchangers, tend to localize at the leading edge of cells, where they could potentially trigger osmotic water flow across the plasma membrane [174].

These ion channels and transporters are implicated in crucial stages of tumor metastasis, such as loss of cell-to-cell contacts, invasion of the surrounding stroma, and entry into and exit from blood vessels (intra- and extra-vasation) [175]. Kourghi and colleagues further deepen this narrative by showing that certain AQP1 ion channel blockers, which do not impact AQP1’s water channel activity, effectively inhibited the migration of HT29 cancer cells. The degree of inhibition was directly related to the potency of the AQP1 ion channel blockage, suggesting that the ion channel properties of AQP1 alone might be sufficient for facilitating tumor cell migration in certain cases [176].

Recent research has also started to focus on the role of AQP1 in the broader tumor microenvironment. Pelagalli and colleagues found that bone marrow-derived mesenchymal stem cells (BM-MSCs)-conditioned medium increased AQP1 expression in U2OS osteosarcoma and SNU-398 hepatocellular carcinoma cells. This upregulation led to enhanced migration and invasion, which could be counteracted by the AQP1 inhibitor, tetraethylammonium chloride [177]. Given that BM-MSCs have been shown to differentiate into cancer-associated fibroblasts that further promote tumor growth and metastasis [178], AQP1 appears to play a significant role in the complex interactions between the tumor microenvironment and tumor cells.

Understanding this multifaceted role of AQP1—encompassing its ion channel properties, its function in osmotic water flow, and its involvement in the tumor microenvironment—could offer valuable insights into new therapeutic avenues for combating cancer metastasis, which is responsible for the vast majority of cancer-related deaths.

#### 7.2.2. AQP1-Modulated Tumor Angiogenesis

Tumor angiogenesis, or the formation of new blood vessels within a tumor, serves as a lifeline for cancer cells, supplying essential nutrients and providing a pathway for metastasis [179]. Ion channels and transporters, including aquaporins such as AQP1, are increasingly being recognized as key players in this process. These proteins serve various roles, including acting as enzymes, chemical and mechanical sensors, receptors, and structural scaffolds [180]. Notably, AQP1 is preferentially upregulated in areas of astrocytomas where tumor cell infiltration occurs, hinting at its potential involvement in tumor angiogenesis [165].

A growing body of evidence further implicates AQP1 in angiogenic processes. Saadoun and colleagues found that mice lacking AQP1 and implanted with melanoma cells showed a reduced density of tumor microvessels, slower tumor growth, and increased survival [181]. This supports previous work that demonstrated diminished tumor microvascular density upon inhibiting AQP1, either through RNA interference or genetic knockouts, in various animal models [182]. Concomitant with this reduction in vascular density was a decrease in markers of angiogenesis, such as the vascular endothelial growth factor receptor 2 (VEGFR2) and the endothelial marker factor VIII [183].

On a cellular level, Saadoun and colleagues showed that endothelial cells deficient in AQP1 had impaired migration and abnormal vessel formation in vitro [181]. Moreover, silencing AQP1 in human endothelial cells (HMEC-1) led to disorganized F-actin polarization at the cell membrane’s leading edge and a failure to establish a cord-like vascular network in culture. Similar findings were also noted in AQP1-silenced human melanoma cells [184]. These data suggest that AQP1 facilitates not just tumor cell migration but also that of endothelial cells, a critical feature for angiogenesis.

The link between AQP1 and endothelial cell migration may also tie into vascular permeability, an early step in angiogenesis. Clapp and Escalera proposed that AQP1 could amplify vessel permeability by enhancing cellular water transport. This, in turn, could trigger an angiogenic cascade by facilitating the extravasation of plasma proteins, which then serve as a scaffold for the migration of endothelial cells [185].

AQP1 appears to have a multifaceted role in cancer progression, impacting both tumor cells and the surrounding microenvironment. Its involvement in angiogenesis adds another layer to its importance, providing further avenues for targeted therapeutic interventions.

#### 7.2.3. AQP1-Modulated Tumor Proliferation

While much attention has been focused on the role of AQP1 in tumor cell migration and angiogenesis, the evidence concerning its role in tumor cell proliferation is less consistent. For instance, inhibition of AQP1 activity had no impact on the proliferation of the colon cancer cell line HT29, whereas a modest 17% reduction was observed in another colon cancer cell line, HCT-116 [186]. Furthermore, overexpression of AQP1 in mouse cancer cell lines B16F10 and 4T1 did not lead to increased proliferation, although it did result in enhanced extravasation and metastases [168]. On the other hand, forced AQP1 expression was linked to heightened cell proliferation in NIH-3T3 mouse embryo fibroblasts and the rat pheochromocytoma cell line PC12 [187].

Adding another layer of complexity is the relationship between AQP1 and resistance to apoptosis. Hoque and colleagues observed that cells expressing AQP1 displayed resistance to apoptosis, possibly contributing to enhanced proliferation [187]. Specifically, AQP1 transfection in PC12 cells was associated with an altered cell cycle profile [188]—increased proportions of cells in the S and G2/M phases and elevated expression of cyclin D1 and E1, which are key proteins for cell cycle progression [189].

Cell volume dynamics are intrinsically tied to cell cycle progression and apoptosis. The cell cycle is marked by an increase in cell volume, while apoptosis involves a reduction in cell volume [190,191]. Given that AQP1-overexpressing PC12 cells exhibited greater cell size and higher intracellular complexity compared to wild-type controls, Galan-Cobo and colleagues hypothesized that these changes in cell morphology might facilitate cell cycle progression and inhibit apoptosis [189].

The role of AQP1 in tumor cell proliferation is not as clearly defined as its roles in tumor cell migration and angiogenesis. While some evidence suggests that AQP1 may contribute to enhanced proliferation and resistance to apoptosis, other studies offer conflicting results. This makes AQP1 an intriguing but complex target for potential therapeutic interventions aimed at inhibiting tumor cell proliferation. Further research is needed to fully understand the mechanisms by which AQP1 may or may not influence this aspect of tumor biology.

### 7.3. Assessing the Prospects of Aquaporin-Targeted Therapies in Malignancies of the CNS

While there is considerable excitement around the idea of aquaporin-targeted therapeutics, especially given promising data from animal studies, progress in this domain has been less than satisfactory [192,193,194]. A number of factors have contributed to this state of affairs, including conflicting reports about the inhibition of AQPs by commonly used ion transport inhibitors such as loop diuretics and antiepileptic medications. The literature is also muddled by reports of small-molecule inhibitors of AQPs that later could not be replicated in follow-up studies.

Given these inconsistencies, there is a critical need for rigorous scientific approaches to evaluating AQP function and inhibitor efficacy. For instance, comprehensive functional screens involving large collections of random, drug-like small molecules could provide valuable insights into effective AQP inhibitors. Alternatively, computational chemistry could guide smaller, more focused screens to identify potent compounds. Either way, robust validation in diverse cellular contexts will be essential to confirm the efficacy of any proposed AQP-targeting compounds.

A relatively unexplored area within this realm is the development of antibody- and peptide-based therapeutics against AQPs. There are also opportunities for research into small-molecule transcriptional regulators that can modulate AQP expression. These innovative approaches could offer alternative ways to target AQPs and their functions, potentially overcoming some of the limitations posed by existing methods and animal models.

Commercial interest in AQP-targeted therapies remains high, and the unmet need for effective treatments for various diseases provides a strong motivation for continued research in this area. Thus, the dual goals of creating viable commercial products and developing better research tools to replace or augment current transgenic animal models are likely to drive further innovations and breakthroughs in the study of AQPs.

## 8. Conclusions

### 8.1. Emerging Therapeutic Interventions and Prospects

Loss-of-function mutations in aquaporins are infrequently associated with human diseases. For example, mutations in AQP2 can lead to non-X-linked nephrogenic diabetes insipidus (NDI), a condition with an extremely low incidence (~1 in 20 million births) [195]. NDI results in severe symptoms such as polyuria and polydipsia that do not respond to antidiuretic hormones. The current treatment primarily involves water replacement and the use of thiazides to reduce urinary water loss. There is also ongoing research into pharmacological chaperone therapies and gene replacement or stem cell treatments as potential therapeutic avenues for NDI related to AQP2 mutations [196].

In contrast to AQP2, very few individuals have been identified with loss-of-function mutations in other AQPs such as AQP1, AQP3, and AQP7. For instance, individuals with AQP1 deficiency, identified through blood-group screenings, generally appear phenotypically normal but exhibit impaired urinary concentrating abilities when water-deprived—similar to AQP1-null mice [197]. Due to the rarity and variability of these conditions, there is limited information available about the roles these AQPs play in human health.

Furthermore, mutations in the major intrinsic protein (MIP, also known as AQP0) have been linked to congenital cataracts [198]. However, recent studies suggest that the primary function of MIP in the lens might be related more to cell–cell adhesion and gap-junction channel regulation rather than water transport.

Overall, the rarity of AQP deficiencies and the variability of their phenotypic expression in humans make it challenging to understand their roles fully. To date, no other disease-causing mutations in AQPs have been described in the medical literature. This highlights the need for continued research to better understand the function of AQPs in human physiology and pathology.

### 8.2. Final Thoughts

The study of aquaporins is an extraordinarily complex field that intersects with a wide range of scientific disciplines, from physiology and biochemistry to cell biology and biophysics. These proteins serve multifaceted roles in various tissues throughout the body, adding layers of intricacy to their understanding.

In this comprehensive review, we have embarked on an intellectual expedition to explore the many dimensions of aquaporins. We delved into their historical development, molecular classification, and the impact they have on neural health, among other areas. We have also scrutinized their involvement in a range of medical conditions, notably within the realms of oncology and neurology. The multi-faceted nature of aquaporins underscores the broad relevance and applicability they possess in the scientific and medical communities.

We earnestly aim for our work to resonate with the pressing issues and objectives of the contemporary world. In doing so, we hope to contribute meaningfully to the search for effective therapies against some of the most devastating diseases facing humanity today. Given the ubiquitous presence of aquaporins throughout the human body, these proteins offer an exciting frontier for therapeutic research, particularly in the quest to address the debilitating impacts of neurodegenerative diseases, acute cerebral events, and cancer.

As we look to the future, one key area that warrants further investigation is the role of aquaporins in conditions such as brain tumors. Given the devastating nature of these malignancies and the still-limited therapeutic options, understanding the impact and regulatory functions of aquaporins in such conditions could yield valuable insights and perhaps even revolutionary treatments.

Overall, while substantial progress has been made in understanding the roles and functions of aquaporins, significant questions remain. As such, future research should not only focus on a deeper understanding of their molecular behavior and physiological impacts but should also explore their potential as therapeutic targets. By pursuing these avenues of inquiry, we are optimistic that we can unlock new, more effective ways to address some of the most intractable health challenges of our time.

## Figures and Tables

**Figure 1 ijms-24-14340-f001:**
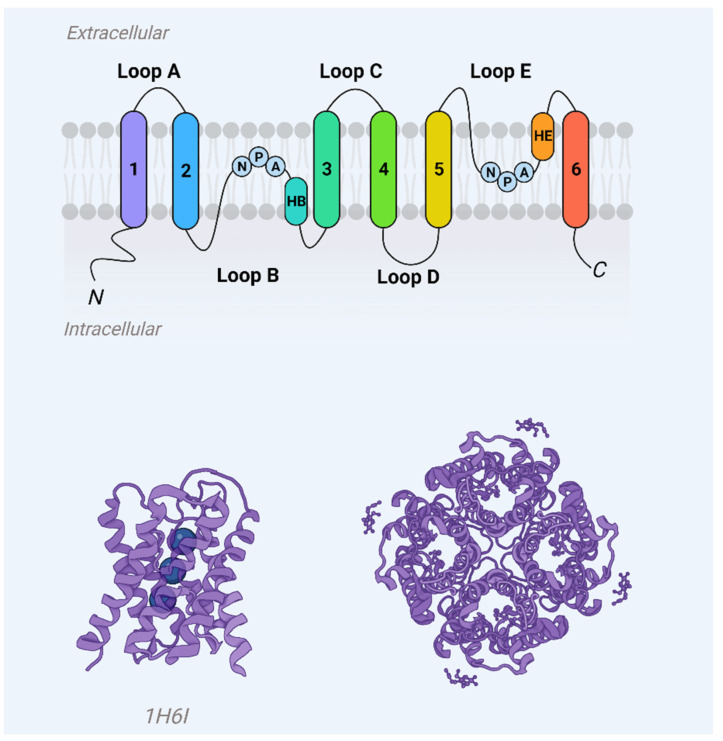
Aquaporin monomer membrane topography (upper panel) and crystal structure (AQP1, PDB 1j4n) showcase helices H1–H8 and water molecules (blue spheres) in the aqueous pore.

**Figure 2 ijms-24-14340-f002:**
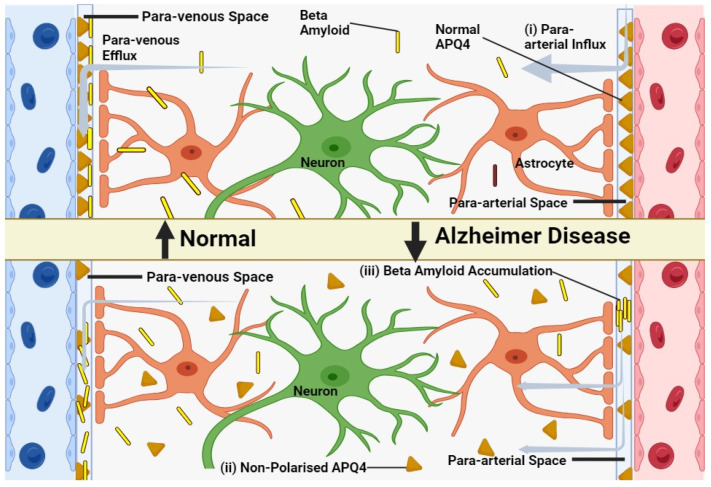
Model of glymphatic function in young, old and in Alzheimer’s disease. (i) CSF efficiently clears brain solutes in young, healthy individuals via periarterial pathways. (ii) Aging disrupts glymphatic function, possibly due to reactive astrocytes and AQP4 de-polarization. (iii) In Alzheimer’s, β-amyloid accumulates in perivascular spaces, potentially due to glymphatic impairment, further hindering waste clearance.

**Figure 3 ijms-24-14340-f003:**
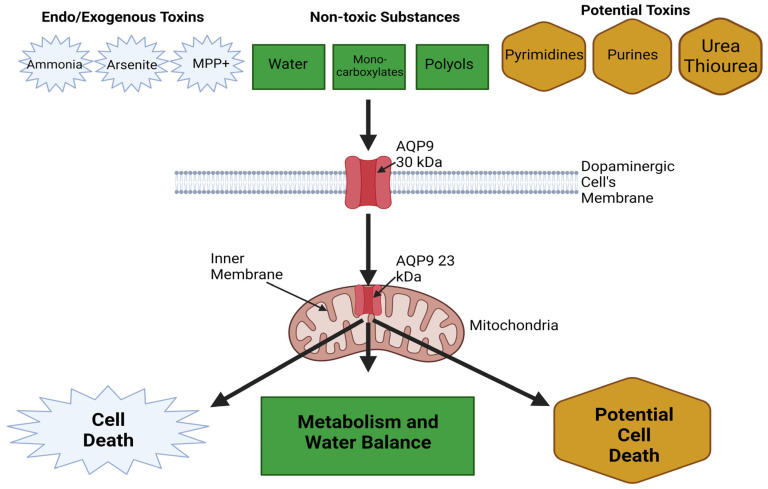
Suggested mechanisms for AQP9-linked dopaminergic cell death and their physiological roles.

**Figure 4 ijms-24-14340-f004:**
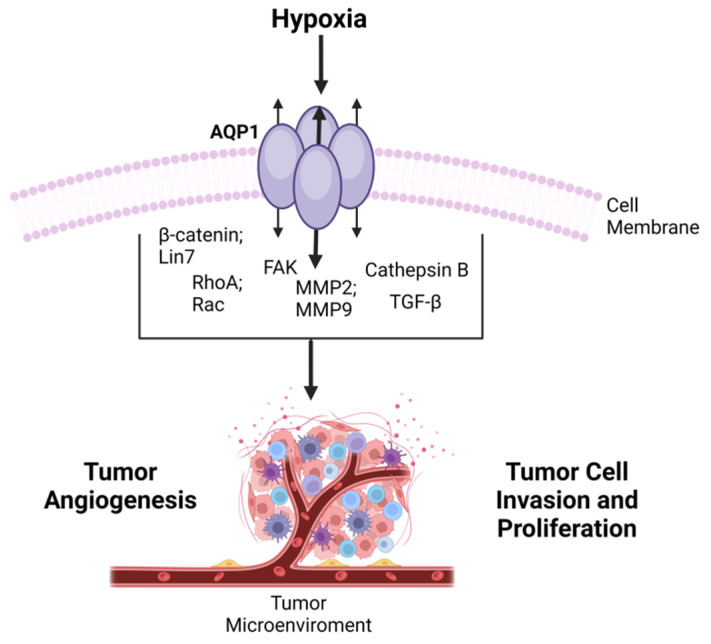
Overview of AQP1 in Cancer Progression-tumor development via transited molecules-emphasizing its involvement in cell migration, invasion, and angiogenesis, and its potential as a prognostic factor in various cancers.

**Table 1 ijms-24-14340-t001:** Regulation of immune-related AQPs during inflammation.

Gene	Species	Immune Cells	Stimuli	Regulation	References
AQP1	Human	Leucocytes	LPS	Upregulation	[144]
	Human	Monocytic THP-1 cells	LPS	Upregulation	[148]
AQP3	Human	Leucocytes	Sepsis	Downregulation	[144]
	Human	Monocytic THP-1 cells	LPS	Upregulation	[147]
AQP5	Human	Monocytic THP-1 cells	LPS	Downregulation	[148]
AQP7	Mouse	Macrophages		Unknown	[150]
AQP9	Human	Leucocytes	SIRS	Upregulation	[149]
	Mouse	Dendritic cells	LPS	Upregulation	[146]
	Human	Macrophages	*Pseudomonas aeruginosa*	Upregulation	[151]
	Human	Leucocytes	LPS	Upregulation	[152]
	Human	Monocytes	LPS	Upregulation	[147]
	Mouse	Macrophages		Unknown	[150]

## Data Availability

Not applicable.

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
