# Peer review of "From Homeostasis to Pathology: Decoding the Multifaceted Impact of Aquaporins in the Central Nervous System"

_ijms, 2023, doi:10.3390/ijms241814340_

Round 1

Reviewer 1 Report

In this paper review entitled “From Homeostasis to Pathology: Decoding the Multifaceted Impact of Aquaporins in the Central Nervous System” of Taodaer et al., the authors provide to make an exhaustive review on the evolutionary history, molecular classification, and physiological relevance of aquaporins, emphasizing their significance in the central nervous system, exploring the role of AQP in neurodegenerative diseases.

First of all, I am pleased for having had the opportunity to read this review very interesting for how he approaches the topic. I also find the historical references on the discovery of AQP very pleasant.

Considering the various mechanisms presented and the various roles that aquaporins play within the central nervous system and in neurodegenerative diseases, more than one graphic scheme or drawing, even a summary one, would be very useful.

I therefore ask the authors to add at least a couple of summary diagrams or drawings for the most important pathological mechanisms in the main body of the review.

Author Response

Dear Reviewer,

Thank you for your positive feedback, appreciation and great suggestions,

We have added more figures and a graphical abstract for better visual representation!

Thank you for your significant contribution!

Reviewer 2 Report

In this manuscript, Toader et al. give a comprehensive review of Aquaporins (AQPs), including the general characteristics of aquaporins, the discovery of AQP1 and significance of aquaporins in cellular physiology. Especially, the authors provide an in-depth exploration of AQP4 and AQP1 in the brain, from their roles in fluid homeostasis to  interplay between AQPs and the glymphatic system, implying a strong association bwtween the dysregulation of AQP-mediated processes in this system with neurodegenerative disorders. 

Some minor suggestions are listed below:

For Figure 1, "Aquaporin monomer membrane topography (left)" should be "Aquaporin monomer membrane topography (upper panel)"

Why water molecules (blue spheres) appear at the central pore of the AQP1 tretramer (lower panel, right)? In addition, please indicate the meaning of the blue chained ball in this figure.

For Figure 2, For clearenss, please label i, ii and iii in the figure; and indicate the symbols, such as AQP4, in the figure legend. 

Author Response

Dear Reviewer,

Thank you for your positive feedback, appreciation and great suggestions,

We have made the modifications to Figure 1 and 2, moreover, we have added 2 more supplementary figures and a graphical abstract for a better visualization!

Thank you for your significant contribution!